# Temperature-Insensitive Refractive Index Sensor with Etched Microstructure Fiber

**DOI:** 10.3390/s19173749

**Published:** 2019-08-30

**Authors:** Bin Dai, Xiang Shen, Xiongwei Hu, Luyun Yang, Haiqing Li, Jinggang Peng, Jinyan Li

**Affiliations:** Wuhan National Laboratory for Optoelectronics, Huazhong University of Science and Technology, Wuhan 430074, China

**Keywords:** optical fiber sensors, fiber optics, refractive index, microstructure fiber

## Abstract

A Mach–Zehnder interferometer (MZI) based on an etched all-solid microstructure fiber (MOF) has been demonstrated. The MZI works on the basis of interference between the vibrant core and cladding modes in the MOF. The all-solid MOF has a heterostructure cladding composed of Ge-doped rod arrays and pure silica, and thus can support and propagate a vibrant cladding mode with a large mode area. When the outermost cladding of MOF is etched, the cladding mode becomes sensitive to the ambient refractive index (RI). The etched MOF can work as a sensing head for RI sensing. By comparing the interference spectra, the extinction ratio has remained stable at around 20 dB after the MOF was etched. The RI sensing characteristics of the MZI with an etched MOF have also been investigated. The results show that the RI sensitivity can reach up to 2183.6 nm/RIU with a low-temperature coefficient (<10 pm/°C).

## 1. Introduction

Since the refractive index (RI) sensors have many applications in biochemical detection and environment tests [1,2], the RI sensors based on optical fibers attract lots of attention [3,4]. Most RI sensors based on fibers take advantage of the evanescent field, and they are sensitive to the changes of the ambient environment. This can be illustrated by fiber interferometers, long period grating (LPG), and fiber Bragg grating (FBG). The LPG can couple light from the core mode to the forwarding propagating cladding mode, which is sensitive to the ambient RI [5]. A Sagnac interferometer based on a high birefringence microfiber can be ultrasensitive to the RI with a sensitivity of 18987 nm/RIU [6]. A Fabry–Perot resonator made by two FBGs on microfiber can be used as an acoustic transducer due to its high sensitivity to the RI [7]. Recently, the Mach–Zehnder interferometer (MZI) has been widely used in the RI sensor for its compact structure and small dimension. An interesting approach in the use of two tapers as an MZI was reported, and the RI sensitivity of MZI is 17.1 nm/RIU [8]. Additionally, the two tapers also can be replaced by a pair of LPGs [9] or special splicing joints [10] to form an MZI. However, an MZI is generally realized by complex and special processing methods, including fiber tapering, dislocation splicing, femtosecond laser micromachining, and excessive discharge/heating. Most of these methods show randomness and lack of precision in modes excitation and coupling processes. The induced structural fragility is also unavoidable. 

The MOFs show great potential in optical fiber sensing applications owing to their adjustable and peculiar properties derived from their various structures. Considerable research efforts have been devoted to MOFs for fiber sensing [11,12]. For example, polarization maintaining a photonic crystal fiber can substitute the Panda polarization-maintaining fiber in a Sagnac interferometer. Thus, the interferometer can be utilized as a twist sensor with temperature insensitivity and lower crosstalk [13]. In addition, with further tapering or special fusing, a microstructure fiber (MOF) also can be used to form MZI. When this kind MZI works as an ultrasensitive RI sensor, the sensitivity can reach 1600 nm/RIU [14]. However, the MOFs with air hole arrays unavoidably exhibit some disadvantages, such as significant splice loss and low mechanical strength. There is thereby an urgent need to design and produce a special MOF as a superior candidate for an MZI-based fiber sensor, but it is still a significant challenge. Our colleagues have proposed and experimentally demonstrated an all-solid MOF with heterostructure cladding [15].

In this paper, the all-solid MOF with heterostructure cladding is sandwiched between single-mode fibers (SMFs) to form a MZI. Different from the traditional MZI, our proposed MZI shows excellent repeatability and convenience in preparation, and can obtain a standard interference spectrum with a high extinction ratio since the MOF used in an MZI can support and propagate a vibrant cladding mode with a large mode area. After the outermost silica is etched out, the MOF becomes sensitive to the ambient environment’s RI, and still exhibits considerable mechanical strength. The proposed MZI shows a RI sensitivity of 2183.6 nm/RIU with temperature independence.

## 2. The Theoretical Analysis and Experiment

Generally, the total internal reflection photonic crystal fiber (TIR-PCF) consists of an arrangement of air holes in the cladding, which has a lower refractive index compared to the fiber core. Therefore, it is difficult for the TIR-PCF to excite and propagate vibrant cladding modes. The splicing loss induced by the air-hole collapse is also adverse to the sensor construction and needs to be eliminated. So, an all-solid photonic band-gap photonic crystal fiber (PBG-PCF) with an arrangement of high refractive index rods is a superior candidate.

According to the anti-resonant reflecting optical waveguide theory [16,17], in the PBG-PCF, the light is mostly scattered back to the fiber core in the condition of phase-matching; on the surface of a high refractive index rod with grazing incidence, there is a strong overlap between the core mode and a leaky rod mode. It is still difficult for a conventional PBG-PCF to achieve the co-existence of a vibrant core and cladding modes. Changing the arrangement of high refractive index rods can increase the confinement loss of the core mode with no obvious change in the band-gap position [18]. So, a microstructurally designed fiber is proposed to increase the confinement loss of the core mode to make sure that the energy can be effectively transferred to the cladding. In the proposed MOF, some of the Ge-doped rods are replaced by pure silica rods, as shown in Figure 1. The heterostructure cladding can be divided to two areas: pure silica and Ge-doped rod arrays. The pure silica area protects the cladding mode from ambient perturbations. The central core and annular Ge-doped rod arrays confine and propagate the fiber modes. According to the evanescent field theory, the evanescent field of the cladding mode gradually becomes stronger as the fiber diameter is reduced. 

The proposed MOF is fabricated in our laboratory [18]. The Ge-doped and pure silica rods are prepared with the diameter of 1 mm. The rods are stacked as shown in Figure 1b. Then, the stacked rods are put into a quartz socket tube to obtain a preform. The preform is fixed in an optical fiber drawing tower to draw fiber. In order to eliminate the air space between the rods, a negative pressure state is applied during the drawing process. The physical parameters of the fiber are described below. The diameter of the fiber cladding and Ge-doped rod arrays is 125 µm and 43 µm, respectively. The diameter of the Ge-doped rod is 1.75 µm, and the pitch is 3.5 µm. The RI of the Ge-doped rod and silica is 1.487 and 1.45, respectively. An MZI is constructed by splicing a piece of MOF between two SMFs (SMF-28). The construction is conducted by a commercial fiber fusion splicer (fsm-60s, Fujikura) with default fusion splice parameters, and thus shows excellent convenience and repeatability. 

The schematic of the experiment is shown in Figure 2a. The lead-in SMF is connected to a broadband light source (SLED, 1250–1650 nm) to input light, and the lead-out SMF is connected to an optical spectrum analyzer (AQ6370D, Yokogawa) to receive a spectral signal. When the input light passes through the splicing joint of a lead-in SMF and MOF, the cladding mode is excited and propagated in MOF. Owing to the effective refractive index differences of the core and cladding modes, there is an apparent phase difference between them, and these two modes interact with each other. According to the Mach–Zehnder interference theory, typical interference fringes can be obtained when these modes recouple into lead-out SMF. The measured transmission spectra of MZI are shown as a black line in Figure 2b. A series of uniform interference fringes with a high extinction ratio (>20 dB) are observed. The insertion loss of splicing an MOF into SMF is mainly due to the mode field mismatch [18]. The transmission intensity of MZI (*I*) can be described as following [19]:
(1)$$I=I_{core}+I_{cladding}+2\sqrt{I_{core}I_{cladding}}\cos(\phi )$$
(2)$$\phi =\frac{2\pi (n_{eff}^{core}-n_{eff}^{cladding})L}{\lambda }$$
where the *I_core_* and *I_cladding_* are the intensities of the core and cladding modes; $n_{eff}^{core}$ and $n_{eff}^{cladding}$ are the effective refractive indexes of the core and cladding modes, and Φ is the phase difference of the core and cladding modes.

When Φ = (2*m* + 1)*π*, *m* = 0,1,2…, the minimum intensity in the interference spectrum can be achieved. From Equations (1) and (2), the free spectral range (FSR) is changed with the length of the MOF. In our previous work [15], the FSRs of an MZI with different MOF lengths are obtained. When the length of the MOF is spliced between SMFs is 22.5 cm, 37 cm, and 45 cm, the corresponding FSR is 26.92 nm, 16.16 nm, and 13.32 nm at around 1550 nm, respectively.

In this experiment, the length of the MOF (*L*) used in our MZI is 17.2 cm. By analyzing the transmission spectrum in Figure 2b, the FSR of the unetched MOF is obtained to be 31.54 nm. To figure out how many cladding modes are involved in the interference, the fast Fourier transform (FFT) of the transmission spectrum is obtained in Figure 3. Only one main peak (0.0325 nm^−1^) is obtained, which means that only one cladding mode mainly interacts with the core mode. At around 1550 nm, the calculated FSR is 30.77 nm, which is close to the experimental result (31.54 nm). It can be initially predicted that the interference mainly comes from two modes. 

In addition, the corresponding effective refractive indexes difference (Δ*n*) of the two involved modes is calculated by Equation (3). The Δ*n* is calculated to be 0.000438 at 1550 nm [19].
(3)$$FSR=\lambda^{2}/\Delta nL$$

The effective refractive indexes of the core and cladding modes at around 1550 nm are calculated by the full-vector finite element method, and the result is shown in Figure 4a [20]. The simulation of the electric field distributions of the core and cladding mode is shown in Figure 4b. The simulated result shows that the Δ*n* is 0.000418 at 1550 nm, which is close to the experimental result (0.000438). Thus, it is further proved that the interference is a typical two-beam interference, and mainly comes from two modes.

The extinction ratio of the interference fringes can reach up to 20 dB and barely decrease, even if the MOF is as long as 0.5 m. According to Equation (1), it is entirely reasonable to predict that the intensity of the two modes involved in Mach–Zehnder interference is roughly comparable. Additionally, the normalized intensity of the electric field in the MOF at 1550 nm is calculated by the full-vector finite element method. From Figure 5, it can be found that the intensity of the core mode approximates to the intensity of the cladding mode. This simulated result coincides with our experimental results. It further proves that this MOF can support and propagate vibrant core and cladding modes simultaneously. It is different from normal multimode fibers where the intensity of the core mode is much higher than that of the other modes. 

The core mode is confined and propagated in the central area of the MOF, while the cladding mode is in the outer periphery of the central core and surrounded by an outermost pure silica cladding. Considering this, it can be predicted that the cladding mode could be more easily influenced by the RI of an ambient environment than the core mode if the outermost pure silica cladding of an MOF is etched out.

To verify this deduction, a series of experiments are conducted. In order to etch out the outermost silica, the MOF is immersed into hydrofluoric acid (HF) solution and subjected to different extent local etching for hours. In this experiment, the concentration of HF solution is 10%. By adjusting the etching time, the fiber diameter of the etched area can be controlled. Since the fiber coating is made by acrylate, which is scarcely etched by HF solution, the length of the etched area also can be controlled by stripping the acrylate. With the same etching length (the stripped length is 4.5 cm), several samples are prepared for comparison. Their diameters of the etched area are 46 μm, 42 μm, and 38 μm, respectively. The transmission spectra of etched and unetched MOFs are measured and shown in Figure 6a. The interference spectra of the etched samples are still distinguishable with negligible etching-induced loss until the diameter of the etched area is decreased to 38 μm. This is because the Ge-doped rod arrays in the heterostructure cladding remain structurally intact during the etching process. Thus, the vibrant cladding mode is still tightly confined and propagated. Specifically, the Ge-doped rod arrays suffer from etching when the diameter is decreased to 38 μm. With the destruction of the heterostructure, the vibrant cladding mode is broadly leaked out, which leads to the interference pattern disappearance and significant leakage loss. Therefore, it is crucial to assure the heterostructural integrity by optimizing the etching time. 

The spectra of etched MOFs and unetched MOF are compared in Figure 2b. When the fiber diameter of the etched area is 46 μm, the corresponding FSR is 26.3 nm. When the fiber diameter of etched area is 42 μm, the corresponding FSR is 22.4 nm. These are both smaller than that of unetched MOF. It also can be seen that the smaller diameter of the etched area leads to narrower FSR. After being etched, Equation (3) can be expressed as Equation (4).
(4)$$FSR=\lambda^{2}/(\Delta n_{1}L_{1}+\Delta n_{2}L_{2})$$

Δ*n*_1_ and Δ*n*_2_ are the effective refractive index difference of the two interactive modes in the etched and unetched areas of the MOF. *L*_1_ and *L*_2_ are the fiber length of the etched and unetched areas. In this experiment, *L*_1_ is 4.5 cm and *L*_2_ is 12.7 cm. The FSR is 21.17 nm at around 1550 nm. The effective refractive index difference in etched area Δ*n*_1_ is calculated to be 0.001357, which is much higher than that of the unetched area (0.000438). 

The analysis of Equation (3) shows that the FSR is in an inverse ratio to the effective refractive index difference between the core and cladding modes (Δ*n*). As the outermost pure silica cladding is being etched out, the intense evanescent field of the vibrant cladding mode will be totally exposed to the ambient environment. As a result, the sample with a smaller etched diameter shows higher sensitivity to the ambient RI. Hence, optimizing the etching time is not only to assure the heterostructural integrity, but also to minimize the diameter of the etched area. Through optimization, an etched sample with 43-um diameter is prepared by immersion into HF solution for about nine hours.

To investigate the RI sensing characteristics of the MZI with an etched MOF, it is submerged into glycerol–water solution, where a refractometer (Reichert, Digital Brix/RI-Chek) is set to obtain a standard value of RI. The RI of the solution can be tuned by changing the percentage of pure glycerol in glycerol–water solution. The corresponding RI can range from 1.3313–1.4468 along with the percentage of glycerol increasing. Before the RI sensing measurement, the RI meter needs to be calibrated. At first, the etched MOF (red line) remains straight with nearly no strain, and is put on a clean and dried slide glass. Then, the etched MOF is carefully cleaned with alcohol and distilled water and air dried. In addition, the etched MOF is also treated with alcohol and distilled water and air dried between tests to avoid any liquid residue.

In the MZI-based RI sensor, the sensitivity can be expressed as follows [21]:(5)$$\Delta \lambda_{m}=\frac{2(\Delta n_{1,m}^{eff}+\Delta n)L}{\left(2m+1\right)}-\frac{2(\Delta n_{1,m}^{eff})L}{\left(2m+1\right)}=\frac{2(\Delta n)L}{\left(2m+1\right)}$$
where the Δ*λ_m_* is the resonance wavelength shift, and *m* is an integer; $\Delta n_{1,m}^{eff}$ is the effective RI difference between the core mode and the m-order cladding mode, and Δ*n* is the change of the effective RI as a result of the change in the ambient RI; and *L* is the length of etched fiber in this paper. It is obvious that the RI sensitivity is proportional to the change of the effective RI (Δ*n*). The spectral response in transmission with different RI is presented in Figure 7a. The interference spectrum undergoes a redshift as the ambient RI increases. The attenuation dip located at 1477.33 nm maintains a complete shape in a large dynamic range, which can be selected as the sensing signal. Figure 7b shows the wavelength shift against the RI changing. For the range of RI from 1.3313–1.4235, the linear sensitivity is 212.07 nm/RIU. As the RI of the solution increases from 1.4335–1.4468, the linear sensitivity has made a rapid and significant increase, and can reach up to 2183.6 nm/RIU. This result is similar with the previously reported works [21,22,23,24]. The reason is that the effective RI of the cladding mode is more sensitive to the ambient RI in the higher RI range [25].

The RI sensitivities of other etched MZIs with different diameters have also been investigated. The results are shown in Figure 8a. When the etched diameter decreases, the attenuation dip comes to be effectively affected by the ambient RI. In addition, the MZI with a smaller etched diameter shows a higher RI sensitivity. It is because the evanescent field of the vibrant cladding mode is gradually exposed to the solution, with the etched diameter reduced to near 42 μm. 

The proposed MZI is placed in a tubular heater to investigate the effects of temperature. The wavelength shift with the temperature changing can be expressed as follows [15]:(6)$$\Delta \lambda =\lambda_{\mathrm{res}}(\frac{1}{\Delta n_{eff}}\frac{\partial \Delta n_{eff}}{\partial T}+\frac{1}{L}\frac{\partial L}{\partial T})\Delta T$$
where *λ_res_* is the resonance wavelength; Δ*n_eff_* is the effective refractive index difference between the core mode and cladding mode with temperature variation; Δ*T* is the temperature variation; and *L* is the length of the MOF. It can be seen that the temperature response mainly comes from two factors: thermal expansion and thermo-optic effects. The thermal expansion coefficient of silica is 5.5 × 10^−7^/K; the thermo-optic coefficients of a Ge-doped rod and pure silica is 8.6 × 10^−6^/°C and 6.9 × 10^−6^/°C, respectively [26]. As reported [15], the thermal expansion coefficient of silica is an order smaller than the effective thermo-optic coefficient of the first cladding, and thus can be neglected. So, the temperature response mainly depends on the thermo-optic effects. The first cladding with a higher effective thermo-optic coefficient consists of a Ge-doped rod and pure silica, where the cladding modes are confined; the core with a lower thermo-optic coefficient consists of pure silica, where the core mode is confined [15]. Therefore, the unetched MOF exhibits similar temperature sensitivity with the etched MOF. The spectral response in transmission with temperature variations is presented in Figure 8b. As the temperature increases from 25 °C to 90 °C, the attenuation dip is shifting only within 0.5 nm with irregularity and poor linearity. The lower temperature sensitivity is a desirable merit for RI sensors to decrease the cross-sensitivity. The sensitivity of temperature is much less than that of RI, which can be neglected in RI sensing [27,28,29]. Therefore, the proposed MZI can be considered as a temperature-insensitive RI sensor.

## 3. Conclusions

In conclusion, a temperature-independent MZI fiber sensor for RI sensing is experimentally demonstrated. Thanks to the heterostructure cladding of the all-solid MOF, a standard MZI interference spectrum with a high extinction ratio can be obtained. The all-solid structure of the fiber also can provide high repeatability and convenience in preparation for MZI. The all-solid MOF offers a certain channel to confine and propagate a vibrant cladding mode. No further complex post-processing is required for the sensor preparation; only adjusting the concentration of HF solution and controlling the etching time is required. Owing to the large mode area of the cladding mode, the MOF needn’t be etched to a small dimension. So, the sensor shows distinctive superiority in mechanical strength and convenience in cleaning. With the outermost pure silica cladding etched out, the maximum RI sensitivity can reach up to about 2183.6 nm/RIU. Meanwhile, this RI sensor is temperature insensitive. Therefore, our proposed MZI is a promising candidate for an RI sensor due to its advantages of convenience in preparation, compact structure, excellent mechanical strength, operating reliability, and repeatability.

## Figures and Tables

**Figure 1 sensors-19-03749-f001:**
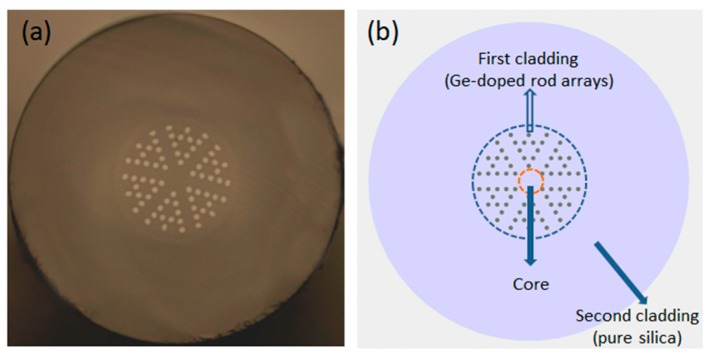
(**a**) Optical micrograph of the cross-section of an all-solid microstructure fiber (MOF); (**b**) Schematic representation of the cross-section.

**Figure 2 sensors-19-03749-f002:**
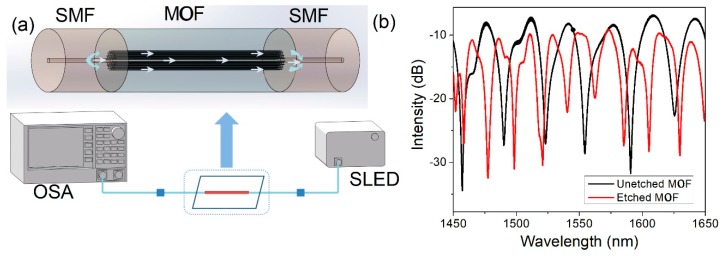
(**a**) Schematic of the experiment; (**b**) The transmission spectrum of unetched MOF and etched MOF.

**Figure 3 sensors-19-03749-f003:**
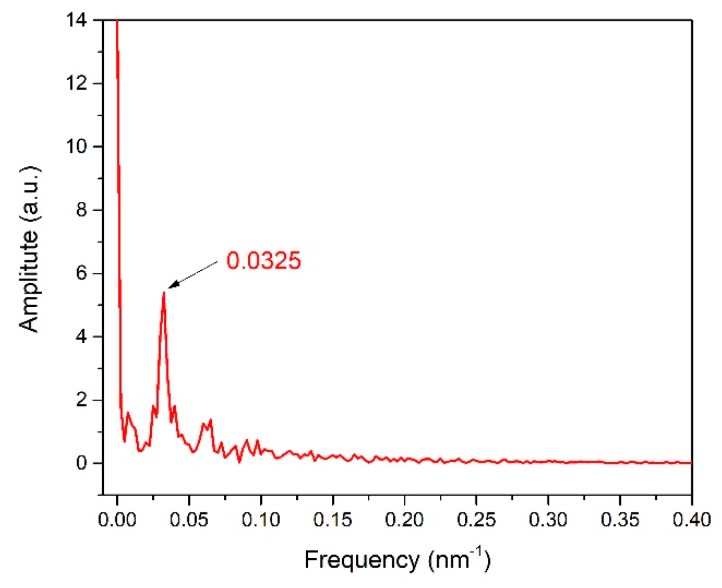
The fast Fourier transform (FFT) spectrum of unetched MOF.

**Figure 4 sensors-19-03749-f004:**
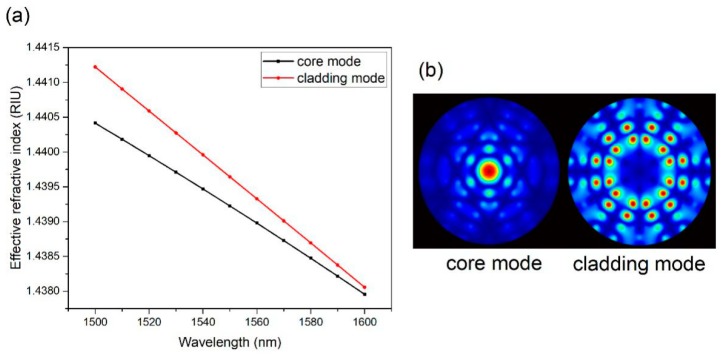
(**a**) The effective refractive indexes of the core and cladding modes; (**b**) The electric distributions of the core and the cladding modes in an MOF at 1550 nm.

**Figure 5 sensors-19-03749-f005:**
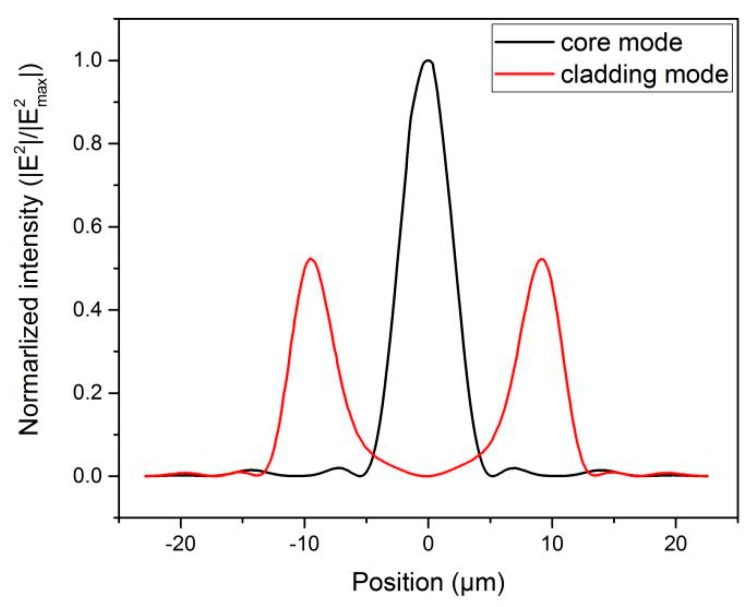
The normalized intensity in the MOF at 1550 nm.

**Figure 6 sensors-19-03749-f006:**
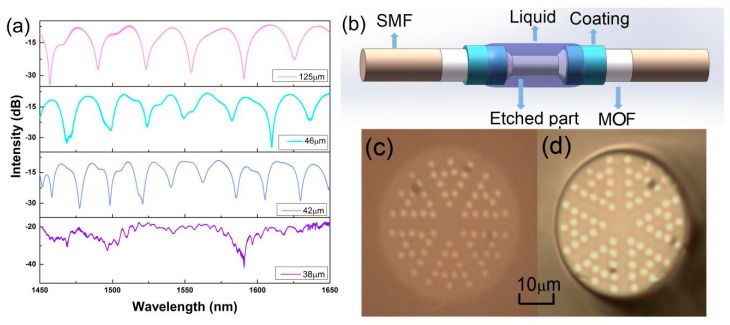
(**a**) The transmission spectra of an unetched MOF and etched MOF with different diameters; (**b**) The schematic diagram of the Mach–Zehnder interferometer (MZI); (**c**) The cross-section of an unetched MOF; (**d**) The cross-section of an etched MOF.

**Figure 7 sensors-19-03749-f007:**
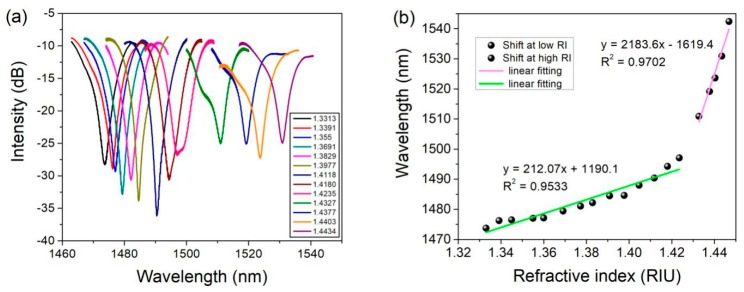
(**a**) The optical spectrum of etched MOF fiber with a diameter of 43 µm; (**b**) The shift of wavelength with the changing of the ambient refractive index (RI).

**Figure 8 sensors-19-03749-f008:**
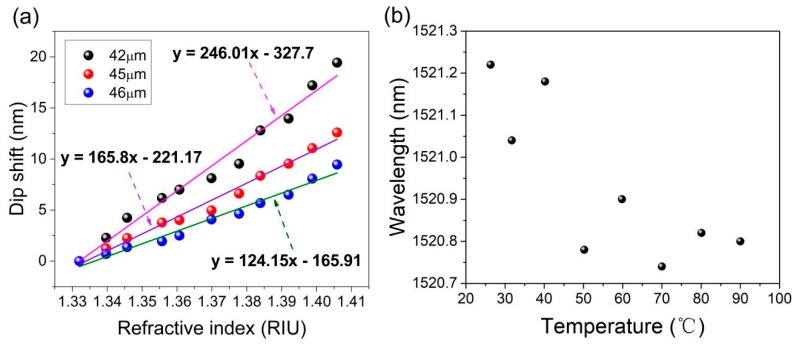
(**a**) The RI sensitivities of the MZI with different etched diameters; (**b**) The temperature response of the RI sensor.

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
