# Peer review of "Temperature-Insensitive Refractive Index Sensor with Etched Microstructure Fiber"

_sensors, 2019, doi:10.3390/s19173749_

Round 1
Reviewer 1 Report
The authors present a temperature independent MZI fiber sensor for RI sensing, which is based on an all-solid microstructure fiber with vibrant core and cladding composed of Ge-doped rod arrays and pure silica. There are still some issues should be further clarified and discussed before the reviewer makes a recommendation for publication in Sensors.
Could the authors provide more information about the fabrication and design criterial of the Ge-doped rod arrays? It there any threshold that need to be considered to make a feasible device? Could authors comments on the insertion loss introduced by splicing the MOF into the SMF? Is there any methodology to improve the loss? In the experimental part on page 6, the RI of solution with pure glycerol was measured. What is the solution used to dilute the pure glycerol? How was the RI sensing measurement was calibrated? Could the author provide the details about calibrating RI meter? Figure 7 presented two different sensitivity, could the authors comment on the reason? Will this limit the measurement range of the device? Figure 8(b) shown the temperature insensitivity of the RI sensor, could the authors explain the reason for this? Is this temperature insensitivity intrinsically happening due to the material? Is this result corresponding to the MOF with 42um etched diameter?
Reviewer 2 Report
after reading the paper I think that it can be accepted after minor revisions
1-please provide references for all equations or report how they are obtained.
2-full-vector finite element method, please give reference or explain it.
3- figure 5, it would be better to report E^2.
4-improve figure 6a quality
5-figure 8a, avoid overlapping between data and equations.
6-how much is the possible error due to the temperature effects reported at the end of the paper
Round 2
Reviewer 1 Report
I recommended the paper to be published in the Sensor in its present form.